

# Multiple, independent colonizations of the Hawaiian Archipelago by the family Dolichopodidae (Diptera)

Kari Roesch Goodman[1], Neal Evenhuis[2], Pavla Bartošová-Sojková[3] and Patrick Michael O'Grady[1]

[1] Department of Environmental Science, Policy and Management, University of California, Berkeley, CA, United States
[2] Department of Natural Sciences, Bernice P. Bishop Museum, Honolulu, HI, United States
[3] Institute of Parasitology, Biology Centre of the Czech Academy of Sciences, České Budějovice, Czech Republic

Corresponding author
Patrick Michael O'Grady,
ogrady@berkeley.edu

## ABSTRACT

The family Dolichopodidae forms two of the four largest evolutionary radiations in the Hawaiian Islands across all flies: *Campsicnemus* (183 spp) and the *Eurynogaster* complex (66 spp). They also include a small radiation of *Conchopus* (6 spp). A handful of other dolichopodid species are native to the islands in singleton lineages or small radiations. This study provides a phylogenetic perspective on the colonization history of the dolichopodid fauna in the islands. We generated a multi-gene data set including representatives from 11 of the 14 endemic Hawaiian dolichopodid genera to examine the history of colonization to the islands, and analyzed it using Bayesian and maximum likelihood phylogenetic methods. We used a subset of the data that included *Conchopus* and the eight genera comprising the *Eurynogaster* complex to estimate the first phylogenetic hypothesis for these endemic groups, then used Beast to estimate their age of arrival to the archipelago. The *Eurynogaster* complex, *Campsicnemus* and *Conchopus* are clearly the result of independent colonizations. The results strongly support the *Eurynogaster* complex as a monophyletic group, and also supports the monophyly of 4 of the 8 described genera within the complex (*Adachia, Arciellia, Uropachys* and *Eurynogaster*). Members of the family Dolichopodidae have been dispersing over vast distances to colonize the Hawaiian Archipelago for millions of years, leading to multiple independent evolutionary diversification events. The *Eurynogaster* complex arrived in the Hawaiian Archipelago 11.8 Ma, well before the arrival of *Campsicnemus* (4.5 Ma), and the even more recent *Conchopus* (1.8 Ma). Data presented here demonstrate that the Hawaiian Dolichopodidae both disperse and diversify easily, a rare combination that lays the groundwork for field studies on the reproductive isolating mechanisms and ecological partitioning of this group.

## INTRODUCTION

Long distance dispersal from continental populations is critical to the formation of the Hawaiian flora and fauna (*Carson & Kaneshiro, 1976*; *O'Grady, Magnacca & Lapoint,*

*2009*), but is considered rare. This infrequent arrival and establishment has led to a flora and fauna that is disharmonic relative to those on the continents that served as sources (*Gillespie & Roderick, 2002*). Recently, several studies (reviewed in *Heaney, 2007*; *Bellemain & Ricklefs, 2008*) have shown that reverse colonization from Hawaii to continental landmasses is observed in birds (*Filardi & Moyle, 2005*), plants (*Harbaugh & Baldwin, 2007*) and insects (*O'Grady & DeSalle, 2008*; *Lapoint, Magnacca & O'Grady, 2014*), suggesting that dispersal plays a larger role than previously thought and evidence is accumulating to indicate that movement to and from island systems is more common, especially at geological time scales (*Heaney, 2007*; *Cibois et al., 2011*; *Hembry et al., 2013*; *Casquet et al., 2015*). If a lineage is vagile enough to repeatedly colonize an area, there is a reduced chance that it will generate the reproductive isolation necessary to speciate and then radiate. Furthermore, if radiation does occur in a lineage and there is subsequent colonization of the area by close relatives, ecological theory would predict that the existing niches would be pre-empted (*Hardin, 1960*), rendering a second radiation unsuccessful. Thus, clear examples where a lineage colonizes and radiates repeatedly and substantially are rare.

The Hawaiian-Emperor Archipelago has a long and dynamic geological history, well isolated in the central Pacific Ocean far from any continental mass. It has been forming by the motion of the Pacific plate over a stationary hotspot (*Wilson, 1963*), generating an island chain that is at least 80 million years old (*Clague & Dalrymple, 1987*; *Duncan & Keller, 2004*; *Sharp & Clague, 2006*). Island formation during this long history has been episodic, with some periods characterized by only few, low elevation atolls and reduced species diversity and other times with multiple high islands capable of supporting a diverse flora and fauna (*Price & Clague, 2002*). Many of the older islands that are now submerged or heavily eroded to small land masses once provided the kind of high island habitat we are familiar with in the contemporary high islands (Niihau, Kauai, Oahu, Molokai, Lanai, Maui, Kahoolawe and Hawaii), which have been forming very recently—only over the past five million years (*Clague & Dalrymple, 1987*; *Clague, 1996*: Fig. 1). The current high islands provide a rich array of habitats, ranging from low to high elevation and very dry to very wet vegetation types.

All of the flora and fauna arrived to this dynamic archipelago via long distance dispersal in an unlikely sequence of events in which taxa both managed to land on the islands and persist once there (*Zimmerman, 2001*; *Gillespie et al., 2012*). Recent phylogenetic studies of Hawaiian insects (*Jordan, Simon & Polhemus, 2003*; *Mendelson & Shaw, 2005*; *Shapiro, Strazanac & Roderick, 2006*; *Medeiros et al., 2009*; *Lapoint, Gidaya & O'Grady, 2011*; *Medeiros & Gillespie, 2011*; *O'Grady et al., 2011*; *Haines & Rubinoff, 2012*; *Bennett & O'Grady, 2013*; *Bess, Catanach & Johnson, 2013*; *Goodman & O'Grady, 2013*; *Lapoint, O'Grady & Whiteman, 2013*; *Goodman et al., 2014*; *Haines, Schmitz & Rubinoff, 2014*; *Lapoint, Magnacca & O'Grady, 2014*), have begun to reveal the history of colonization to and diversification within the Hawaiian Archipelago, and it appears that history is somewhat idiosyncratic. Some large groups, such as Hawaiian Drosophilidae with an estimated 1,000 species, colonized the Hawaiian Islands tens of millions of years ago. Other diverse groups, such as *Nesophrosyne* leafhoppers, with 72 described and over 100 undescribed species (*Bennett & O'Grady, 2011*), and *Campsicnemus* flies with about

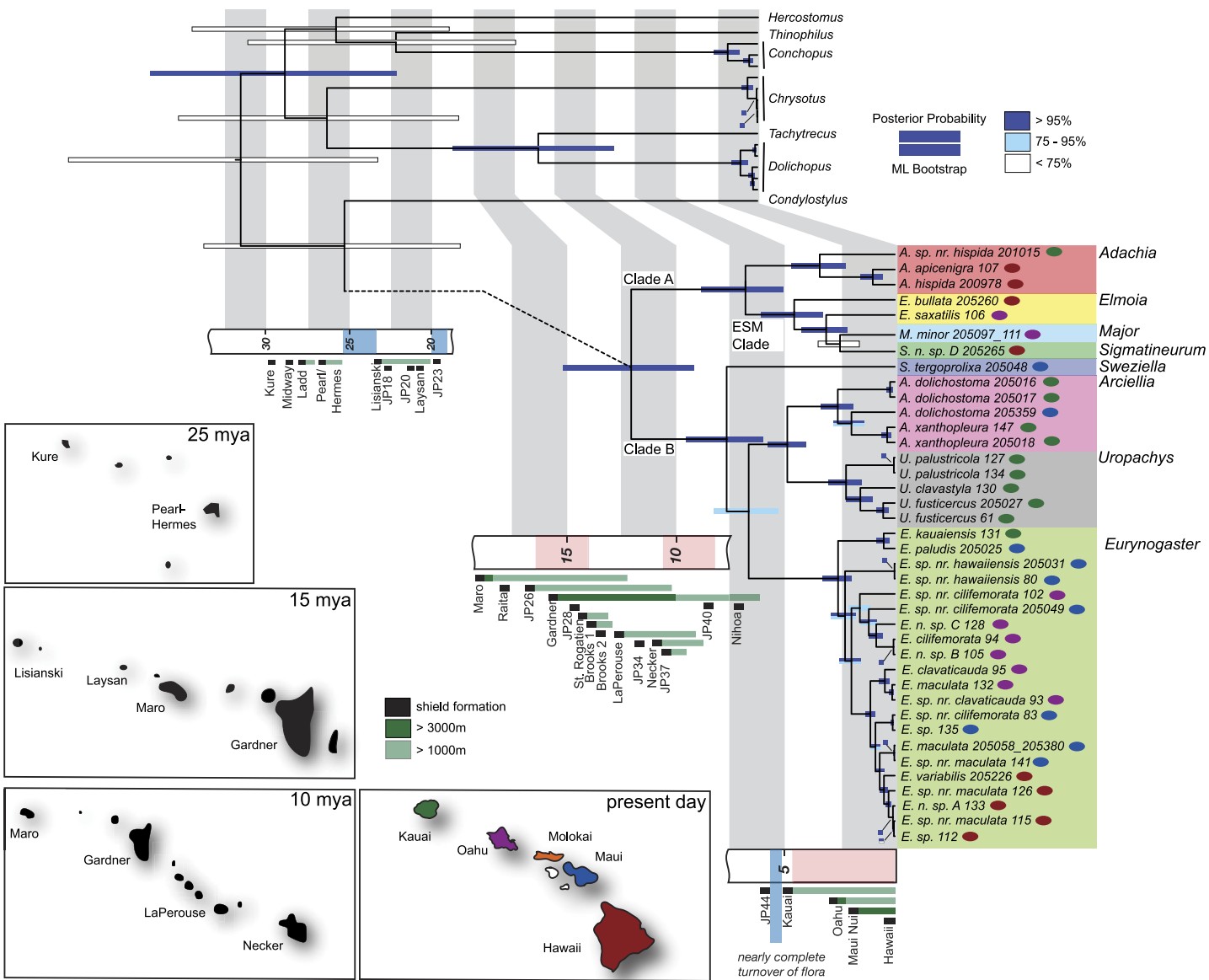

**Figure 1** **Maximum clade credibility tree summarizing BEAST analysis of the *Eurynogaster* complex with geologic history of the archipelago.** Node bars are the 95% highest posterior density intervals of the divergence time estimate. The color of each bar indicates the level of maximum likelihood bootstrap support for each clade (dark blue, >95%; light blue, 75–95%; white, <75%). Timeline of shield formation (black), maximum elevation > 3,000 m (dark green) and maximum elevation > 1,000 m (light green) is shown for the past 30 million years. Island area estimates, redrawn from *Price & Clague (2002)*, are shown for the present day and 10, 15 and 25 million years before present. Island color in the present day corresponds to island occurrence for each sampled taxon (red, Hawaii; blue, Maui; orange, Molokai; purple, Oahu; green, Kauai).

200 species (*Goodman et al., 2014*) are young, dating to only a few million years. One thing is clear, however—very few endemic Hawaiian plant or animal families have successfully colonized the islands multiple times (e.g., Araliaceae; *Plunkett, Soltis & Soltis, 1997*; *Costello & Motley, 2001*) and in no case have any of these generated two radiations of with more than 50 species each.

**Table 1** **Composition and status of Dolichopodidae fauna of Hawaii.** Genera with endemic species are **boldface**.

| Genus | Total spp. in Hawaii | Number of endemic spp. | Number of non-endemic spp. | Number of described spp, included in this study (undescribed spp.) [included from outside Hawaii] |
|---|---|---|---|---|
| *Achradocera* | 2 | 0 | 2 | 0 |
| *Amblypsilopus* | 1 | 0 | 1 | 0 |
| *Asyndetus* | 1 | 1 | 0 | 0 |
| *Austrosciapus* | 1 | 0 | 1 | 0 |
| *Campsicnemus* | 183 | 183 | 0 | 70[14] |
| *Chrysosoma* | 2 | 0 | 2 | 0 |
| *Chrysotus* | 1 | 0 | 1 | 1(1)[1] |
| *Conchopus* | 6 | 6 | 0 | 3 |
| *Condylostylus* | 1 | 0 | 1 | 1 |
| *Dactylomyia* | 1 | 0 | 1 | 0 |
| *Diaphorus* | 1 | 0 | 1 | 0 |
| *Dolichopus* | 1 | 0 | 1 | 1 |
| *Eurynogaster complex* | | | | |
| *Adachia* | 6 | 6 | 0 | 2 (1) |
| *Arciellia* | 3 | 3 | 0 | 2 |
| *Elmoia* | 8 | 8 | 0 | 2 |
| *Eurynogaster* | 23 | 23 | 0 | 6 (7) |
| *Major* | 1 | 1 | 0 | 1 |
| *Sigmatineurum* | 11 | 11 | 0 | 1 |
| *Sweziella* | 7 | 7 | 0 | 1 |
| *Uropachys* | 7 | 7 | 0 | 3 |
| *Hydrophorus* | 2 | 2 | 0 | 0 |
| *Krakatauia* | 1 | 0 | 1 | 0 |
| *Medetera* | 1 | 0 | 1 | 0 |
| *Paraliancalus* | 2 | 2 | 0 | 0 |
| *Pelastoneurus* | 1 | 0 | 1 | 0 |
| *Sympycnus* | 1 | 0 | 1 | 1[5] |
| *Syntormon* | 1 | 0 | 1 | 1[5] |
| *Tachytrechus* | 1 | 0 | 1 | 1 |
| *Thinophilus* | 1 | 1 | 0 | 1 |

Flies in the family Dolichopodidae are remarkable in that they have colonized the Hawaiian Islands multiple times and still have managed to generate two of the largest evolutionary radiations within the Hawaiian Diptera: *Campsicnemus* Haliday, 183 spp. (*Goodman et al., 2014*), and the *Eurynogaster* complex, 66 spp. in eight genera (*Evenhuis, 2005*). In addition, they also generated a small radiation of 6 spp., *Conchopus* Takagi. In addition to these three radiations, four other dolichopodid genera contain endemic species: *Asyndetus* (1), *Hydrophorus* (2), *Paraliancalus* (2), and *Thinophilus* (1) (Table 1). Thus, the family Dolichopodidae offers a unique opportunity to examine the timing and frequency of long distance colonization events in the founding of the endemic Hawaiian fauna. While recent molecular phylogenies of Dolichopodidae (e.g., *Lim et*

*al., 2010*; *Bernasconi, Pollet & Ward, 2007*) have sampled some of these genera (e.g., *Campsicnemus, Hydrophorus, Thinophilus*), uneven sampling between studies and the lack of Hawaiian exemplars makes it difficult to infer the colonization history in detail. Furthermore, while the biogeography of *Campsicnemus* has been studied (*Goodman et al., 2014*), the evolutionary relationships among the three radiations and the monophyly and biogeography of the large *Eurynogaster* complex have never been examined.

The primary goal of this paper is to address the colonization history of the endemic Hawaiian Dolichopodidae and assess how many colonization events have generated the present-day diversity within this lineage. We sampled 11 of the 14 genera with endemic Hawaiian species and included samples from across the family Dolichopodidae. We sequenced a combination of five mitochondrial and two nuclear genes and used these data to estimate colonization times using the Bayesian algorithm implemented in BEAST to infer the colonization history of this family in Hawaii. With our sampling we also provide the first molecular phylogenetic analysis of the *Eurynogaster* complex, with which we assess the monophyly of this lineages and its constituent genera.

## MATERIALS & METHODS

### Taxonomic sampling

Specimens were collected from 2004 to 2012 from sites across the Hawaiian Islands. The bulk of Hawaiian Dolichopodidae species are endemic to high elevation (900–1,700 m.) rain forest habitats, and thus collecting efforts were concentrated in these areas. Other habitats (e.g., coastal strand, dry and mesic forests, alpine zone) were also sampled, including rocky beaches, the only known habitat of *Conchopus, Thinophilus, Asyndetus* and *Hydrophorus*. We succeeded in collecting specimens from 11 of the 14 Hawaiian dolichopodid genera with endemic species known from the islands (*Campsicnemus, Conchopus, Thinophilus* and eight genera from the *Eurynogaster* complex, Table S1A in Appendix S1). Data from the Hawaiian *Campsicnemus* are included here from a previous study from our group, and are described in Appendix A from *Goodman et al. (2014)*. Material was collected by general sweeping of vegetation and leaf litter, pan and Malaise trapping, and hand collecting. To evaluate monophyly of and diversity within the *Eurynogaster* complex, we included representatives from each of its eight constituent genera (Table 1; *Evenhuis, 2005*). No *Eurynogaster* complex lineages were omitted from our sampling. All material was preserved in 95% ethanol.

All material was identified using the most recent key to species in *Tenorio (1969)* and *Evenhuis (2005)*. Descriptions of new species from within the *Eurynogaster* complex discovered as a result of this project are in preparation. Unpublished new species included in the study were given letters (e.g., *Eurynogaster* n. sp. A, B, C, etc.). In addition to the extracted specimens, whenever possible, a series of conspecifics from the same site were also preserved in 95% ethanol. Voucher material has been deposited in the Bernice Pauahi Bishop Museum (Honolulu). In addition, new sequences were generated for outgroup specimens from the non-endemic Dolichopodidae: five specimens of *Dolichopus exsul*, two specimens of *Chrysotus longipalpis*, and one specimen each of *Condylostylus sp.* and

**Table 2  Primer names and references.** Mitochondrial primer numbers correspond to the location in the *Drosophila yakuba* mitochondrial genome (*Clary & Wolstenholme, 1985*). Sequences with no reference were designed as a part of this study.

| Primer name | Length | Genome | Reference or Sequence |
| --- | --- | --- | --- |
| Cytochrome Oxidase I (COI): 2183 or 2640 and 3041 | 829 | mitochondrial | *Bonacum et al. (2001)* |
| Cytochrome Oxidase II (COII): 3037 and 3771 | 681 | mitochondrial | *Bonacum et al. (2001)* |
| NADH Dehydrogenase 2 (ND2): 192 and 732 | 527 | mitochondrial | *Bonacum et al. (2001)* |
| 16S | 530 | mitochondrial | *DeSalle (1992)* |
| 12S | 559 | mitochondrial | F14233, R14922 (*Simon et al., 1994*) 12S_exF: 5′-TCC AGT ACA TCT ACT ATG TTA CG-3′ 12S_inF: 5′-ATG TGT RCA TAT TTT AGA GC-3′ 12S_inR: 5′-TAT TRG CTA AAT TTG TGC CAG C-3′ |
| rudimentary (CAD), nested reaction: 320F and 843R, 338F and 680R | 896 | nuclear | *Moulton & Weigmann (2004)* |
| EF1αA | 1,036 | nuclear | EF4 and EF5 (*Collins & Wiegmann, 2002*) EFF: 5′-CNC CTG GCC ATC GTG ATT TC-3′ EFR: 5′-CAG CAT CTC CYG ATT TGA TGG C-3′ |
| EF1αB | 858 | nuclear | EFF_B: 5′-GAT TAC TGG TAC ATC TCA AGC-3′ EFR_B: 5′-TAG CAG CAT CYC CYG ATT-3′ |

*Tachytrechus angustipennis.* Finally, sequences from *Hercostomus indonesianus* were also downloaded from GenBank to include in the outgroup (see Table S1A in Appendix S1). Access and collection permits were granted by the State of Hawaii Department of Land and Natural Resources, the National Park Service (Hawaii Volcanoes and Haleakala National Parks), Maui Land and Pineapple, East Maui Irrigation, Parker Ranch, and The Nature Conservancy of Hawaii (Appendix S4).

## Phylogenetic analysis
### Relationships within Dolichopodidae and colonization of the Hawaiian Islands

To address the question of whether the endemic dolichopodid fauna, including the three major radiations (*Campsicnemus*, the *Eurynogaster* complex and *Conchopus*) is the result of a single or multiple colonizations, new sequences were generated for the samples described above (and in Table S1A in Appendix S1) and were combined with the entire data matrix generated from the *Goodman et al. (2014) Campsicnemus* study. Genomic DNA was extracted from individuals using a Qiagen DNeasy (Qiagen Inc.) DNA extraction kit, following the manufacturer's protocol. Loci used are described in Table 2. Thermal cycling involved a simple protocol for EF1a, a touchdown protocol for the mitochondrial genes and a nested reaction for CAD (described in *Moulton & Weigmann, 2004*). The simple protocol began with an initial denaturing step at 95C for 4 min, 30 cycles of 90C for 30 s, 54–58C for 30 s, 72C for 60 s and a final extension for 5–10 min 72C. The touchdown protocol began with an initial activation cycle at 96C for 2.5 min followed by 25 cycles of 30 s denaturing at 96C, 30 s annealing through a touchdown series starting from 55C and stepping down 0.4C per cycle, with 45s extension at 72C. This was followed by 15 cycles of 30 s denaturing at 96C, 30 s annealing at 45C and 45 s extension at 72C. A final

extension for 7 min at 72C ended the touchdown protocol. PCR products were purified using Exo-SAP-IT (USB Corporation, Cleveland, OH, USA) following standard protocols, and the products were sent to the UC Berkeley DNA Sequencing Center for sequencing in both directions on an ABI 3,730 capillary sequencer. Eleven of the 14 dolichopodid genera with endemic species are represented. This yielded an alignment, referred to as dataset A, containing 183 individuals and seven loci containing 4,763 base pairs that was used to assess deep temporal and biogeographic patterns within Hawaiian Dolichopodidae.

### Phylogenetic relationships within the Eurynogaster complex

To assess the monophyly of the *Eurynogaster* complex and its component genera, seventeen described, four new, and five possible new species (labeled as "sp. nr".) were included in the phylogenetic analysis (Table 1). This matrix was designated as dataset B. Phylogenetic analyses were performed on a data set consisting of 57 individuals (see Table S1 in Appendix S1) and seven loci containing 5,908 base pairs. The more restricted taxon sampling in dataset B was to maximize the completeness of the seven loci sampled, many of which weren't sampled in the larger dataset A. Results between the two studies are largely congruent. Analyses were conducted on each gene individually using maximum likelihood (ML, see below). Dataset B was used to assess biogeographic patterns within the *Eurynogaster* complex of genera.

Datasets A and B were both analysed using ML and Bayesian inference (BI) optimality criteria. For each of the ML and the BI analyses, the optimum partitioning schemes were calculated in PartitionFinder (*Lanfear et al., 2012*). The optimal partitioning scheme for the combined analysis of Hawaiian Dolichopodidae (dataset A), was calculated from 18 original data partitions (16S, 12S and 1st, 2nd, and 3rd codon positions for COI, COII, ND2, CAD, EF1α and one CAD intron region). Partitioning was calculated for the *Eurynogaster* complex dataset (dataset B) from 20 original data partitions (16S, 12S and 1st, 2nd, and 3rd codon positions for COI, COII, ND2, CAD, EF1αA and EF1αB, intron regions for CAD, EF1αA, EF1αB and ND2) and selected using Bayesian Information Criterion (Table S2B in Appendix 2). For both datasets, in the BI analyses, the best-fit model of sequence evolution for each data partition was also selected using PartitionFinder (Table S2B in Appendix S2: *Lanfear et al., 2012*). Selection of models and partitions proceeded as described above and these are reported in Table S1B in Appendix S1. The ML analyses were performed on individual genes and on the concatenated data sets in RAxML 3.7.2 (*Stamatakis, 2006*) on CIPRES (*Miller, Pfeiffer & Schwartz, 2010*) under the GTR GAMMA model with 1,000 bootstrap replicates and a final search for the best tree. The BI analyses were performed on the concatenated data sets using MrBayes 3.1.2 (*Huelsenbeck & Ronquist, 2001*) on CIPRES (*Miller, Pfeiffer & Schwartz, 2010*), with each analyses run for 30 million generations with 2 independent runs each.

*MCMC convergence diagnostics*: For the BI analyses, stationarity was assessed within and convergence among each of the runs using several complimentary approaches: (1) convergence metrics provided by MrBayes 3.1.2 were checked (*Huelsenbeck & Ronquist, 2001*) to ensure that the maximum standard deviation of split frequencies of any of the runs was under 0.05 and that the potential scale reduction factor for all parameters approached

1.0, and (2) the log-likelihood values for each run were plotted, the Effective Sample Sizes (ESS) were checked to ensure there were an adequate number of independent samples, and the posterior distributions of all parameters were examined using Tracer v.1.72 (*Rambaut & Drummond, 2012*). Tracer v.1.72 was also used to determine the burn-in phase by assessing each run's plot of log-likelihood values over generations—stationarity was assumed to have been reached when the log likelihood values reached a stable plateau. Finally, a 50% majority rule consensus trees was created from the resulting post burn-in trees.

## Divergence time estimation in the *Eurynogaster* complex

To estimate the age of the *Eurynogaster* complex lineage, divergence time estimation was performed on dataset B using a Bayesian relaxed-clock method implemented in BEAST 1.7.5 (*Drummond et al., 2012*) on CIPRES (http://www.phylo.org: *Miller, Pfeiffer & Schwartz, 2010*). The age of the *Eurynogaster* complex is unknown as representatives of the genus are not known outside of Hawaii and biota in the Hawaiian Islands does not fossilize well. There is a fossil available for one genus that has an endemic species in the Hawaiian Islands (*Thinophilus* Wahlberg: subfamily Hydrophorinae), but the wide range in ages of the fossils (Baltic amber—Eocene/Oligocene; ca. 35–60 mya) compared with the very young ages of the islands make them unsuitable for use in this analysis. Instead, we used three biogeographic calibrations based on the island ages of Kauai, Maui and Hawaii (see Table S2A and Fig. S2A). We then evaluated the impact of the Kauai calibration by running an analysis with only the Maui and Hawaii calibrations, and also ran two alternate analyses for comparison based on evolutionary rates, described in Appendix S2.

We selected two well-supported nodes for calibration from within a lineage of the genus *Eurynogaster* that exhibit a clear progression from older to younger islands (Oahu to Maui to Hawaii). We also performed a maximum likelihood ancestral state reconstruction in MESQUITE v.2.7.2 (*Maddison & Maddison, 2009*) to assign ancestral areas to all nodes in the phylogeny. We then selected a third well-supported node for calibration with a clear ancestral range reconstruction to the oldest island of Kauai. All three nodes were calibrated with island dates from *Carson & Clague (1995)* (Table S2A and Fig. S2A). While island calibrations have been widely used for the estimation of divergence times in Hawaiian lineages (e.g., *Rubinoff & Schmitz, 2010*; *Lerner et al., 2011*), it is plausible that divergence among populations occurred prior to island emergence and was thus unrelated, or that it occurred well after the emergence of the younger island (*Heads, 2005*). Standard deviations were chosen to accommodate some of this uncertainty, including a biologically relevant timeframe during which habitat was likely available on the islands, and the fact that the insects may have colonized the islands well before or after they reached their peak heights (Table S2A).

Divergence time estimation was performed on dataset B described above. The same seven gene concatenated data set (COI, COII, ND2, 12S, 16S, EF1$\alpha$ and CAD) was analysed in each of the analyses described here and in Appendix S2. Partitions and the best fit models of evolution for each partition were selected using BIC in PARTITIONFINDER (*Lanfear et al., 2012*). Initial analyses indicated that these models overparameterized the data in that the ESS values were extremely low for some parameters, despite being run

with very long chains (beast-users Google group discussions). For the final runs, all GTR models were changed to HKY (Table S1B) and ESS increased significantly while divergence times and tree topology did not change. Base frequencies were estimated from the data. The partitioning scheme in the divergence rate analyses differed only slightly from the island calibration analyses in that COI was assigned its own partition (Table S1B). Site and clock models were unlinked and all partitions were analysed using an uncorrelated lognormal relaxed clock except for the partition comprised of CAD (positions 1 & 2) and the EF1α intron, for which a strict clock could not be rejected and was thus applied. The tree-shape prior was linked across partitions and the tree-shape prior was specified as a Yule Process. The xml file was hand edited to include a starting tree, generated using maximum likelihood in RAxML 3.7.2 (*Stamatakis, 2006*). Two independent MCMC searches were conducted, each running for 50 million generations and sampled every 1000 generations. The number of generations was selected to generate ESS values greater than 200 for each of the parameters (*Drummond et al., 2012*). Convergence was assessed using TRACER v. 1.7.5 and trees were summarized to one Maximum Clade Credibility (MCC) tree using TREE ANNOTATOR v. 1.7.5 after removing a burn-in phase.

## RESULTS & DISCUSSION

### Phylogenetic relationships within the endemic Hawaiian Dolichopodidae

The family Dolichopodidae includes more than 6,800 described species (*Yang et al., 2006*) in 232 genera worldwide (*Pape & Thompson, 2013*). A total of 29 genera are found in the Hawaiian Islands. Of these, fifteen have been introduced in the past 150 years, most likely through human activity, while the remaining fourteen genera present in the archipelago are known to contain endemic Hawaiian taxa (Table 1). The relationships between *Campsicnemus* and the *Eurynogaster* complex and the colonization history of these genera have remained an open question, largely due to the difficulty of placing both in a subfamilial context. While *Campsicnemus* is clearly placed in the subfamily Sympycninae, the placement of the *Eurynogaster* complex has been more difficult to ascertain (see Appendix S3). Individual taxa have previously been described as members of the subfamilies Sympycninae, Hydrophorinae, and Thinophilinae. *Hardy & Kohn (1964)* considered *Eurynogaster* and associated genera as part of the Sympycninae (see Fig. S3A in Appendix 3). Later, *Evenhuis (2005)* transferred the entire *Eurynogaster* complex to the Hydrophorinae. If the current taxonomy placing these lineages in two separate subfamilies is correct, *Campsicnemus* and the *Eurynogaster* complex represent independent colonizations to the Hawaiian Islands.

Molecular evidence demonstrates that the endemic Hawaiian dolichopodid fauna is clearly the result of multiple colonizations to the archipelago (Fig. 2, Figs. S1A & S1B in Appendix S1). Several key nodes are well supported and allow us to infer the history of the Hawaiian Dolichopodidae. *Conchopus* (posterior probability (PP) = 1, bootstrap (BS) = 100: node A, Fig. 2), the *Eurynogaster* complex (PP = 1, BS = 100: node B, Fig. 2), and *Campsicnemus* (PP = 1, BS = 98: node C, Fig. 2) are each supported as monophyletic

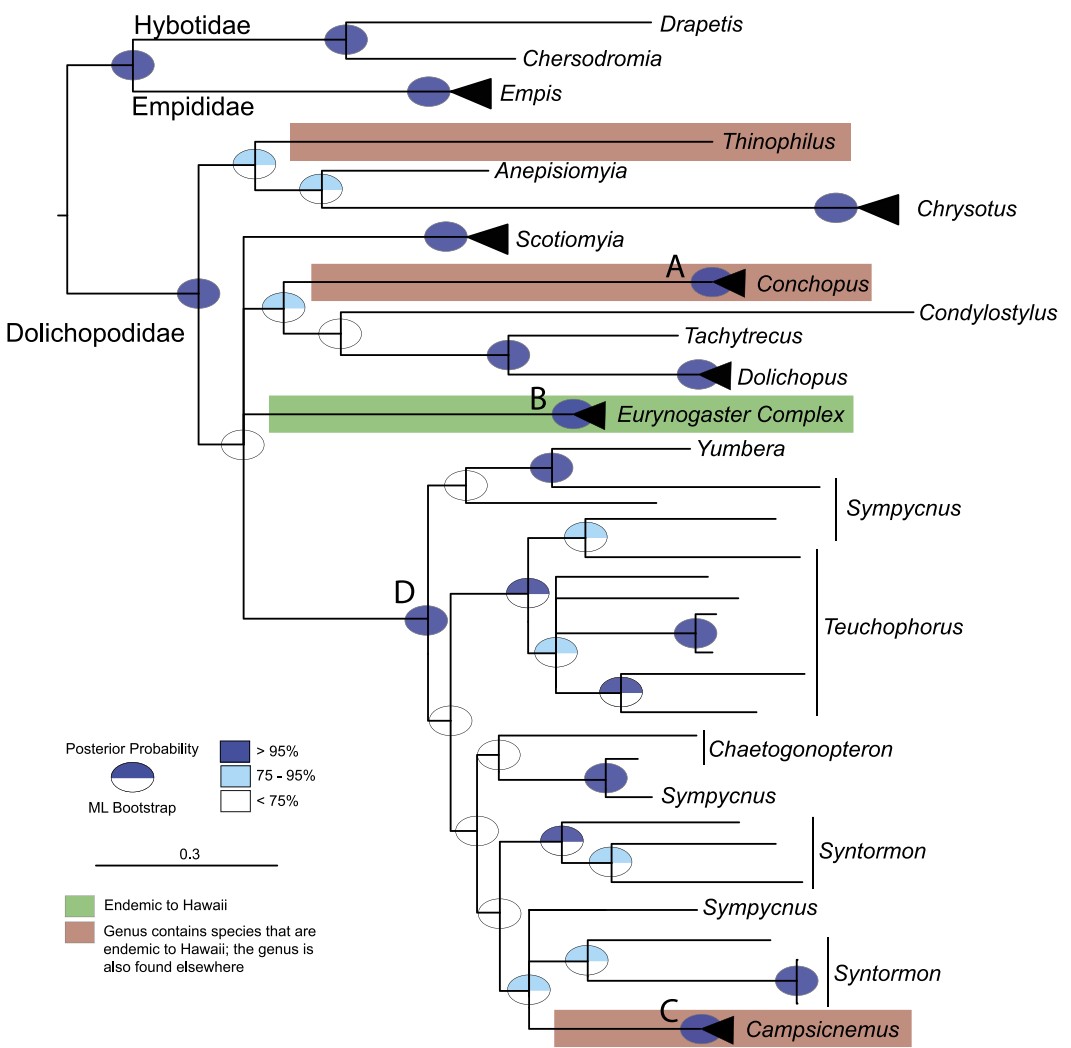

**Figure 2 Majority rule consensus tree summarizing Bayesian analysis of the endemic Dolichopodidae, with the large radiations, *Eurynogaster* complex and *Campsicnemus* collapsed.** Bayesian posterior probabilities (Mr. Bayes) and bootstrap supports from the maximum likelihood analysis (RAxML) are displayed as ovals.

with respect to other dolichopodid genera. Another key node that is supported in both analyses (PP = 1, BS = 99: node D, Fig. 2) is the large clade that includes *Campsicnemus* and a number of non-Hawaiian genera in the subfamily Sympycninae (e.g., *Sympycnus*, *Teuchophorus*) and does not include the *Eurynogaster* complex. This demonstrates that there were at least three colonizations to Hawaii by the family Dolichopodidae, one each by the three radiations: *Campsicnemus*, the *Eurynogaster* complex, and *Conchopus*.

There is little support for the placement of *Thinophilus*, so its history of arrival to Hawaii remains enigmatic (Fig. 2, Figs. S1A & S1B in Appendix S1). This genus is known primarily from the Indo-Pacific, with one species each known from the Galapagos Islands and the Hawaiian Islands. Previously it has only been collected from rocky, wet sand on the south shores of Oahu (*Carlton & Eldredge, 2009*). The specimen included in this study represents

the first record from Hawaii Island and suggests that directed collecting on the south shores of Maui and Kauai may turn up additional populations.

Three genera that contain endemic taxa were not included in this study because they are difficult to collect and we did not recover them in our sampling. While their placement must await future work, their omission here does not change the result that the Hawaiian Islands have been colonized multiple times. An additional issue is that support at many nodes in this phylogeny is poor, owing partially to the large divergences between the subfamilies and the incomplete taxon sampling with this enormous family. These issues are also seen in previously published phylogenetic studies of dolichopodid relationships (*Lim et al., 2010*; *Bernasconi, Pollet & Ward, 2007*). The lack of support and long branches across most of the rest of this phylogeny preclude identifying the specific sister lineages to the Hawaiian taxa (Fig. 2).

## Phylogenetic relationships in the *Eurynogaster* complex

The *Eurynogaster* complex, with 66 described species (*Yang et al., 2006*) and about a dozen awaiting description, comprises the fourth most species-rich radiation of Hawaiian flies, after the Drosophilidae, *Campsicnemus,* and *Lispocephala* (Muscidae). Although the *Eurynogaster* complex is one of the largest radiations of Diptera in Hawaii, phylogenetic relationships in this group have never been studied. This collection of genera are hypothesized to have been derived from a single colonization to the Hawaiian Archipelago (*Evenhuis, 2005*). Little is known about the biology of these species, but collecting observations suggest that species found on the forest floor and on vegetation tend to be dull coloured, while species found in wet habitat, along seeps, streams and on wet banks tend to have shiny metallic thoraces and/or abdomens.

Molecular phylogenetic results presented here show support for *Evenhuis*'s (*2005*) hypothesis of a monophyletic complex of related genera (PP = 1, BS = 100: Fig. 2), as well as support for several of the genera within this radiation. We focused on the smaller dataset (dataset B) to address phylogenetic and biogeographic questions within the *Eurynogaster* genus complex. Analyses of individual genes are presented in Figs. S1E–S1L in Appendix S1, and final data partitions and evolutionary models are reported in Table S1B in Appendix S1. Tree topologies generated using ML and BI approaches of the concatenated dataset B were very similar; at well-supported nodes, they are identical (Figs. S1C & S1D).

In Fig. 1, the maximum clade credibility tree from the Bayesian analysis performed in BEAST is used to display the patterns within the *Eurynogaster* complex, and the following PP and BS supports are from the BI performed in MrBayes and ML analysis performed in RAxML (shown in Figs. S1C & S1D). The *Eurynogaster* complex is split into two clades: Clade A (*Adachia* + *Elmoia* + *Sigmatineurum* + *Major*, PP = 1, BS = 100) and Clade B (*Sweziella* + *Arciellia* + *Uropachys* + *Eurynogaster*, PP = 1, BS = 100). Current sampling indicates that the genus *Adachia* is monophyletic (PP = 1, BS = 100) and sister to a well-supported clade (PP = 1, BS = 100) composed of the genera *Elmoia, Sigmatineurum* and *Major* (ESM Clade). Sampling within the ESM clade is not extensive, with only a single representative each of *Sigmatineurum* and *Major*. Two representatives of the genus

*Elmoia* were sampled and our results indicate that this genus is paraphyletic with respect to *Sigmatineurum* and *Major*. Denser sampling with the ESM clade will be necessary to resolve the placement of the *Elmoia* taxa.

Clade B includes the large genus *Eurynogaster*, along with *Arciellia*, *Uropachys* and *Sweziella*. *Sweziella*, represented by *S. tergoprolixa* from Maui, is the basal lineage within clade B and sister to the lineage formed by *Arciellia, Uropachys* and *Eurynogaster* (PP = 1, BS = 91: Fig. 2). Current sampling indicates that the genus *Arciellia* and *Uropachys* are each monophyletic (PP = 1, BS = 100 and PP = 1, BS = 100, respectively) and sister to one another (PP = 1, BS = 100). *Eurynogaster* is supported as monophyletic (PP = 1, BS = 100). This genus is confusing taxonomically and is in need of revision. There are three undescribed *Eurynogaster* species that were discovered as part of this work, *E. n. spp. A–C*. There are also a number of taxa that, while morphologically similar to named taxa, show significant sequence divergence from the described species. This sometimes corresponds to samples having been taken from different islands. For example, *E. maculata* from Oahu is quite different from the *E. sp. nr. maculata* samples collected from Maui (*E. sp. nr. maculata 141*) and Hawaii Island (*E. sp. nr. maculata 115* and *126*)–they are 3.9% and 3.6% divergent at COI, respectively. Furthermore, one exemplar of *E. maculata* from Maui is quite similar to *E. sp. nr. maculata 141*—it is identical at COI–suggesting that cryptic species may exist within the concept of what we currently recognize as *E. maculata*. This phenomenon is common in large evolutionary radiations in Hawaii (e.g., *Bennett & O'Grady, 2011*). Another species we sampled, *E. cilifemorata*, also seems to be a complex of species sampled from Maui and Oahu. Additional sampling within *Eurynogaster*, as well as thorough taxonomic revisions of the genera within this complex, will be necessary to better delineate species within this rapidly evolving clade.

Finally, four new species within the *Eurynogaster* complex were discovered as a result of this project, three within *Eurynogaster* and one within *Sigmatineurum*. An additional five possible new species (*Adachia*—1 species; *Eurynogaster*—4 species) were identified (labeled as "sp. nr".) and are in the process of examination to confirm their taxonomic status.

## Arrival times and biogeography

We estimate that the *Eurynogaster* complex arrived in the Hawaiian Archipelago 11.83 (9.08–15.04) Ma, approximately within the timeframe that the Northwest Hawaiian Islands of La Perouse, Necker, and Gardner were providing substantial high island habitat (*Price & Clague, 2002*). This ancient lineage arrived well before the formation of the current high islands about 5 Ma and the arrival of *Campsicnemus*, which is estimated to have occurred approximately 4.6 Ma (*Goodman et al., 2014*). Early diversification into five of the eight contemporary genera took place in the older, now eroded, northwest Hawaiian Islands, and five colonizations of these ancestral lineages into the current main (high) islands are needed to explain the contemporary patterns of diversity. All of the diversification within the crown groups has occurred within the past 5 million years (Myr), the timeframe of the current high islands. The most speciose lineage within the *Eurynogaster* complex, the genus *Eurynogaster*, began diversifying approximately 2.6

(95% HPD: 1.94–3.26) Ma, about the time Oahu and Maui Nui were forming. We estimate that the small endemic dolichopodid genus *Conchopus* arrived quite recently—1.77 (95% HPD: 1.09–2.6) Ma (Fig. 1).

Within the *Eurynogaster* complex, a number of classic biogeographic patterns are evident, some of which are significantly different from what is observed in other large radiations. First, a progression rule pattern (*Hennig, 1966*) is common in hotspot archipelagos where islands appear along a chronosequence. The typical progression rule pattern seen in Hawaii occurs when the most basally branching taxon is present on Kauai, the oldest island, with more recently branching taxa present on the progressively younger islands of Oahu, Molokai, Maui and Hawaii (*Wagner & Funk, 1995*). While the progression rule is commonly observed in both the Hawaiian *Drosophila* (*Bonacum et al., 2005*) and *Campsicnemus* (*Goodman et al., 2014*) lineages, it is less prevalent in *Eurynogaster*. Only a single lineage of the genus *Eurynogaster* shows a clear progression from Oahu to Maui to Hawaii (Fig. 1).

Another phenomenon observed in Hawaiian lineages is within-island diversification, where species break up to diversify into new populations and eventually sibling species on the same island. This has been thought to be an uncommon occurrence, in part because it is fairly uncommon across the historically best-studied group in the islands, the Hawaiian *Drosophila*—for whom diversification primarily occurs following inter-island dispersal. However, even within this iconic group, there are examples and it has been very well studied in the sympatric sibling pair *D. silvestris* and *D. heteroneura* (*Carson, 1982*; *DeSalle et al., 1987*; *Price & Boake, 1995*). Newer examples are now accumulating across taxonomic groups (e.g., *Goodman, Welter & Roderick, 2012*; *Eldon et al., 2013*; *Bennett & O'Grady, 2013*; *Liebherr, 2015*), exposing how variable a process diversification can be, and how dependent it is on the dispersal capabilities of the groups studied (*Price & Wagner, 2004*). The *Eurynogaster* complex shows at least five instances of within-island diversification. *Uropachys* is a genus of six species only known from Kauai. Three *Uropachys* species were sampled for this study and are supported as a monophyletic clade, indicating they diversified there. This pattern is also observed in *Adachia*, where *A. hispida* and *A. apicenigra* have both formed on Hawaii, and in several clades of the genus *Eurynogaster* where diversification has occurred on Oahu and Hawaii. While it is possible that additional sampling, and subsequent discovery of new species, may alter this inference, it is clear that diversification within an island is a pattern seen in many other Hawaiian groups, including the genus *Campsicnemus* (*Goodman et al., 2014*) and the well-studied Hawaiian *Drosophila* (*O'Grady et al., 2011*).

## Colonization of and diversification within the Hawaiian Islands

It is clear that the endemic Dolichopodidae of Hawaii arrived to the archipelago in at least three successful colonization to radiation sequences over the last 12 Myr (*Eurynogaster* complex, 11.8 Ma; *Campsicnemus*, 4.6 Ma; *Conchopus*, 1.8 Ma) –demonstrating that dispersal to and establishment within this remote island group is more common than has been documented in other groups. This is fascinating because it means that three separate radiations occurred despite the excellent dispersal capabilities of these animals. In order
to multiply into radiations, they must have been able to generate reproductive isolation rapidly enough to overcome gene flow from their highly vagile conspecifics. Members of this family are known to have complicated courtship behavior (*Zimmer, Diestelhorst & Lunau, 2003*). Though this has never been studied in the Hawaiian fauna, it may be a contributing factor to the development of reproductive isolation as has been shown with the Hawaiian *Drosophilidae* (*Kaneshiro, 1976*; *Price & Boake, 1995*), *Laupala* (*Grace & Shaw, 2011*) and *Nesosydne* (*Goodman et al., 2015*), and suggests fruitful research directions. There seems to be no correlation between the age of colonization and the diversity of each lineage.

*MacArthur & Wilson (1967)* stated that "an island is closed to a particular species when the species is excluded… by competitors already in residence…" The Hawaiian Islands were clearly not closed to dolichopodid flies that arrived after the first wave 12 Ma. This suggests that, at the arrival of each new lineage, there was still plenty of ecological opportunity available or these insects are ecologically labile and able to adapt easily when faced with niches already occupied by competitors. Both statements may be true.

Very little is known about the ecology of the Dolichopodidae in Hawaii, but they are known to be predatory from observations elsewhere in the world (*Ulrich, 2005*). For the *Eurynogaster* complex (which only occur in Hawaii), there is only a single published account in the literature that includes ecological observations (*Williams, 1938*). Despite the dearth of ecological data available, we have some evidence to support the idea that the dolichopodids seem to adapt easily. In our 2014 study, we used morphological colouring together with field observations to infer that the Hawaiian *Campsicnemus* have rapidly diversified into three ecological types: (1) brown, low vegetation and litter dwellers, (2) black water skaters and (3) yellow canopy dwellers. Interestingly, the black water skaters and yellow canopy dwellers are restricted to the Pacific. Furthermore, the yellow canopy dwellers are endemic to Hawaii (*Goodman et al., 2014*). The Hawaiian *Conchopus* may have also undergone a shift in ecological type. This lineage can be traced back to East Asia (*Takagi, 1965*), where they are known primarily from barnacle colonies in the marine tidal zone, living in the interstices or in nearby cracks in the rocks and feeding on tiny invertebrates (*Sunose & Sato, 1994*). There are no native barnacles in the Hawaiian Islands, and *Conchopus* there are known from *puka* (holes) in beach rocks deriving from volcanic flows. Once established in this habitat, they radiated into six known species.

Prevailing dogma among Hawaiian evolutionary biologists in the past 30 years has been that colonization events to the archipelago are rare and colonization within the islands follow a few well-defined patterns, such as the progression rule (*Wagner & Funk, 1995*). Recent molecular phylogenetic studies are beginning to overturn these overly simplified notions (*Heaney, 2007*; *Bellemain & Ricklefs, 2008*), finding that colonization and diversification are based on a combination of factors. These include characters linked to the dispersal and adaptability of the lineage in question and the ecological and environmental context of the islands when that lineage arrives. The current study highlighting the multiple colonizations that Dolicohpodidae have undergone in the past, and the specific patterns of diversification within the *Eurynogaster* complex, further demonstrate that there are no simple "rules" and each colonization event should be considered an independent

event. The Hawaiian Dolichopodidae are an intriguing example of repeated, overlapping evolutionary radiations, ripe for field studies that can begin to untangle their propensity to speciate and ecological lability.

## ACKNOWLEDGEMENTS

The following people are thanked for their contribution to this project: Bob Peck, Karl Magnacca, Gordon Bennett, and Dan Polhemus for material; Sue Wang, Saya Wai and Crystal Teng for help in the lab, and Dan Bickel and David Hembry for comments on the manuscript. We also thank the State of Hawaii Department of Land and Natural Resources, the National Park Service, Maui Land and Pineapple, East Maui Irrigation, Parker Ranch, and The Nature Conservancy of Hawaii for access and permission to collect. This paper constitutes Contribution No. 2016-019 to the Hawaii Biological Survey.

### Funding

This work was funded by National Science Foundation Grant DEB 0842348 to PMO. The funders had no role in study design, data collection and analysis, decision to publish, or preparation of the manuscript.

### Grant Disclosures

The following grant information was disclosed by the authors:
National Science Foundation: DEB 0842348.

### Competing Interests

The authors declare there are no competing interests.

### Author Contributions

- Kari Roesch Goodman conceived and designed the experiments, performed the experiments, analyzed the data, contributed reagents/materials/analysis tools, wrote the paper, prepared figures and/or tables, reviewed drafts of the paper.
- Neal Evenhuis contributed reagents/materials/analysis tools, wrote the paper, reviewed drafts of the paper.
- Pavla Bartošová-Sojková performed the experiments, reviewed drafts of the paper.
- Patrick Michael O'Grady conceived and designed the experiments, analyzed the data, contributed reagents/materials/analysis tools, wrote the paper, prepared figures and/or tables, reviewed drafts of the paper.

### Field Study Permissions

The following information was supplied relating to field study approvals (i.e., approving body and any reference numbers):

We received access and permission to collect from the State of Hawaii Department of Land and Natural Resources, the National Park Service, Maui Land and Pineapple, East Maui Irrigation, Parker Ranch, and The Nature Conservancy of Hawaii. The permits can be found in Appendix S4.

## DNA Deposition

The following information was supplied regarding the deposition of DNA sequences:

All DNA sequences generated in this study have been deposited in Genbank (Appendix S1).

## Data Availability

The raw data has been supplied as a Supplemental File.

## Supplemental Information

Supplemental information for this article can be found online at http://dx.doi.org/10.7717/peerj.2704#supplemental-information.

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
