# Peer review of "Multiple, independent colonizations of the Hawaiian Archipelago by the family Dolichopodidae (Diptera)"

_PeerJ, doi:10.7717/peerj.2704_

## Round 0.1 · original submission · Minor Revisions

This paper is well-written and will be a good addition to the literature after the reviewers' various points are addressed adequately.

All of the reviewers make good points, and I particularly point to reviewer#2, point #2 (dating analysis) as a point that needs to be addressed in detail in the rebuttal.

I look forward to a revised MS along with a point-by-point response/rebuttal to the reviews. Thank you for your submission to PeerJ.

·

Basic reporting

This paper presents a mostly sound analysis and preliminary assessment of some interesting and valuable questions concerning Hawaii’s endemic dolichopodid fauna. It’s among the first serious attempts to place this large radiation(s?) in a phylogenetic context, and does a pretty good job.

Experimental design

As the authors acknowledge, the sampling outside the Hawaiian fauna is very limited, which limits their conclusions to an extent. I found this a little surprising, given the ease of collecting members of this group everywhere that the authors are. For the most part the results provide adequate caveats, and the main conclusion that, regardless, multiple colonizations are required to account for the Hawaiian endemics isn’t affected. But it wouldn’t have taken all that much more outgroup sampling to be able to say at lot more, it seems. I’d be a particularly cautious about accepting the ‘strong support’ for the monophyly of the Eurynogaster complex, on these grounds, particularly given their ambiguous placement to subfamily (and consequent uncertainty just what some appropriate outgroups might be).

My other primary concern is that the authors don’t address or justify the reasons for their separate analysis of relationships within the ‘Eurynogaster complex’. It may implicitly indicate that there were issues with resolution or completeness in the larger (183 taxon) analysis? Maybe not, but then why wouldn’t the single analysis serve all their purposes adequately? I see that the loci sequenced are not all identical, but it seems that there’s more than enough overlap to unite in a single matrix, without having serious concern about missing data. I wouldn’t necessarily say they need to go back and combine these, but they should make it clearer to the reader what purpose the two separate analyses are serving. The fact that all results are presented as ‘summary’ trees, also clouds the issue just what was included in each.

Validity of the findings

The section on within-island diversification (the claim that they’ve demonstrated at least ‘five clear instances’) seems a little overconfident to me, again based on sampling, in this case of Hawaiian taxa. Confidently inferring such would demand comprehensive sampling, and knowing beyond doubt that, for example, Uropachys occurs on no other island. In each case, discovery of an unsampled, related species on another island could affect the inference. That the authors weren’t even able to sample all the endemic genera, let alone species, and discovered numerous new species in the process, makes it clear that there's still a lot of sampling to be done.

Reviewer 2 ·

Basic reporting

The supplementary material is very important for the manuscript and should be included in the main paper. The only exceptions are Figures S2B-S2D because the main results are also in the supplementary tables.

It is not a good idea that important elements of the methods are described in another paper: “Extraction, amplification, sequencing, editing and alignment followed the same protocols described in Goodman et al. (2014)” At least editing and alignment belong here.

Experimental design

no comment

Validity of the findings

In this manuscript, the authors analyze genetic data for 11 of the 14 Hawaiian dolichopodid genera in order to answer some pressing questions: how often and when did Dolichopodidae colonize the Hawaiian Islands? The answer to the first question requires phylogenetic information while the answer to the second question requires a solid dating analysis based on a well-supported tree. Overall, I would argue that there is sufficient data for demonstrating multiple colonizations, but I am not convinced that there is sufficient information for a dating analysis. Overall, the most convincing part of the manuscript is the phylogenetic analysis of the Eurynogaster complex.
(1) The authors use five mitochondrial and two nuclear markers for reconstructing the phylogenetic relationships. These markers work well in the analysis of the relationships within the Eurynogaster complex. Many clades are well supported in both analyses (ML and Bayesian). However, the same markers yield a poorly supported tree for Dolichopodidae. This is not surprising given that the reconstruction is based on a very small number of species and genes for a taxon that is very old and species-rich. Overall, one can only consider those nodes as being reasonably well supported that have high ML support (dark-blue on Figure 2; PP supports tend to be inflated). There are three such nodes that are above the genus level and lie within Dolichopodidae. I agree with the authors that these nodes can be used to argue for at least 3 colonization events: (1) Eurynogaster complex, (2) Campsicnemus and (3) Conchopus, Thinophilus. The latter may turn out to be two origins once better supported trees become available.
(2) However, I don’t think it is justified to run a dating analysis based on the tree in Figure 2. The node support is too poor and the taxon sampling is too sparse. Many of the nodes are likely to change. In addition, I find two of the three calibration points unconvincing (Figure S2A). They are based on biogeographic arguments. However, figure S2A illustrates that most clades have species that are on 2 of the 4 islands. This means that – despite partial sampling of the species – a fair amount of dispersal has to be assumed which makes it questionable whether ancestral areas can be reconstructed with sufficient certainty. I can still understand why the authors considered the calibration point for Hawaii convincing, but the one for Maui is questionable because the clade includes species from Maui and Oahu. Similarly, the Kauai calibration point is for a clade whose species are found in Kauai and Maui. Surely, there is so much dispersal that the confidence in the ancestral state assignment is not high. This affects the reliability of the dating analysis.
(3) The supplementary material is very important for the manuscript and should be included in the main paper. The only exceptions are Figures S2B-S2D because the main results are also in the supplementary tables.
(4) Minor points:
Why do the authors use “unlikely” in “All of the flora and fauna arrived to this dynamic archipelago via long distance dispersal in an unlikely sequence of events in which taxa both managed to land on the islands and persist once there”

Additional comments

see above

·

Basic reporting

Well written

Experimental design

Rigorously analyzed

Validity of the findings

Well supported

Additional comments

This is a very good manuscript. I have only the most minor recommendations indicated by posted notes on the pdf version of the manuscript.

---

## Round 0.2 · accepted · Accept

The authors have responded adequately and in detail to the various comments and concerns raised by the reviewers, which has improved the MS. It is now acceptable for publication. Thank you for submitting your work to PeerJ.